Journal of Machine Learning Research 25 (2025) 1-12       Submitted 1/25; Revised 5/25; Published 9/25

# Chromosome Mask-Conditioned Generative Inpainting for Atypical Mitosis Classification

**Sweta Banerjee**               SWETA.BANERJEE@HS-FLENSBURG.DE
*Flensburg University of Applied Sciences, Flensburg, Germany*

**Viktoria Weiss**
*University of Veterinary Medicine, Vienna, Austria*

**Thomas Conrad**
*Freie Universität Berlin, Berlin, Germany*

**Taryn A. Donovan**
*The Schwarzman Animal Medical Center, New York, USA*

**Jonas Ammeling**
*Technische Hochschule Ingolstadt, Ingolstadt, Germany*

**Rutger H.J. Fick**
*Diffusely, Paris, France*

**Jonas Utz**
*Friedrich-Alexander-Universität Erlangen-Nürnberg, Erlangen, Germany*

**Robert Klopfleisch**
*Freie Universität Berlin, Berlin, Germany*

**Christopher Kaltenecker**
*Medical University of Vienna, Vienna, Austria*

**Christof A. Bertram**
*University of Veterinary Medicine, Vienna, Austria*

**Katharina Breininger**
*Julius-Maximilians-Universität Würzburg, Würzburg, Germany*

**Marc Aubreville**
*Flensburg University of Applied Sciences, Flensburg, Germany*

**Editor:** My editor

## Abstract

Atypical mitoses are critical prognostic markers for tumor proliferation, yet classification efforts are compromised by class imbalance, data scarcity, and noisy labels. Our work focuses on hematoxylin and eosin (H&E)-stained histopathology images, where identifying such mitoses is particularly challenging due to overlapping morphological features and stain variability. We address these challenges with a novel approach for biologically informed inpainting, conditioned on a histological context patch, an inpainting mask, and a chromosome segmentation mask. This triple-conditioned generative strategy allows disentanglement of the mitotic figure shape information from the cellular context and enables the utilization of large-scale datasets that do not contain atypical sub-classification for training classification models. We evaluate both adversarial and denoising diffusion-based inpainting strategies.

Our approach mitigates the lack of data diversity and label noise, thereby substantially improving classification performance for atypical vs. normal mitoses - as demonstrated by downstream classification with EfficientNet-B0 and Low-rank adaptation (LoRA) fine-tuned foundation models. We provide the complete source code, including all our methods, at our github repository: `https://github.com/DeepMicroscopy/ChroMa-GI`.

**Keywords:** Atypical Mitoses, Generative Inpainting, Classification

## 1 Introduction

Tumor proliferation is an important prognostic indicator in cancer, as highly proliferative tumors often exhibit more aggressive behavior and poorer patient outcomes (Bertram et al., 2024). In routine histological sections, proliferating (i.e., dividing) tumor cells can be identified as mitotic figures (MFs). MFs can be classified into two types – normal and atypical. Atypical mitotic figures (AMFs), characterized by irregular or aberrant chromosomal arrangements (Donovan et al., 2021), have recently been linked to aggressive tumor behavior. Their potential as independent prognostic factors has been demonstrated in breast cancer (Lashen et al., 2022; Ohashi et al., 2018) and other tumors (Jahanifar et al., 2023; Bertram et al., 2023; Matsuda et al., 2016). Yet, AMFs remain underexplored in deep learning studies. In particular, the low prevalence of AMFs and disagreement among experts have limited the development and validation of robust models and hindered their incorporation into large prognostic cohorts.

MFs exhibit high intra-class variance, as they transition through five continuous and morphologically overlapping phases: prophase, prometaphase, metaphase, anaphase, and telophase (Donovan et al., 2021). This variability complicates both detection and classification. AMFs add an additional layer of complexity, as they differ from normal MFs through subtle or pronounced morphological deviations. These may include multipolarity, polar asymmetry, lagging chromosomes, i.e., chromosomes not in contact with the larger chromosomal aggregate, or fine chromosomal connections across spindle poles (bridging) (Donovan et al., 2021). While there is a high intra-class variance of AMFs, the inter-class variance between normal MFs and the AMFs may be small especially for subtle atypical alterations.

These variations result in disagreement among experts when labeling AMFs, leading to significant inter-rater variability and potential label noise in annotated datasets (Bertram et al., 2025). Because of this, dual-stage pipelines are preferable: the first stage detects all MFs, and the second stage classifies image patches cropped around these detections (Fick et al., 2024) into normal vs. atypical MFs. This second stage enables class balancing and supports more advanced regularization and augmentation strategies. In this work, we therefore focus exclusively on the task of MF subclassification (atypical vs. normal), which can be seamlessly integrated as a second stage into any state-of-the-art MF detection system. As shown in prior research (Bertram et al., 2025), this subclassification task remains challenging for current deep learning models, mainly due to inter-rater disagreements and imbalanced datasets.

In this paper, we propose a novel framework that leverages generative models to enhance data diversity and improve AMF classification. Image inpainting, i.e., the generation of missing image segments based on surrounding context, has been explored before for cell images (Kropp et al., 2024; Lu et al., 2019; Oh and Jeong, 2023; Zhang et al., 2020), e.g., to overcome dataset imbalance (Oh and Jeong, 2023; Öttl et al., 2024), or for artifact removal

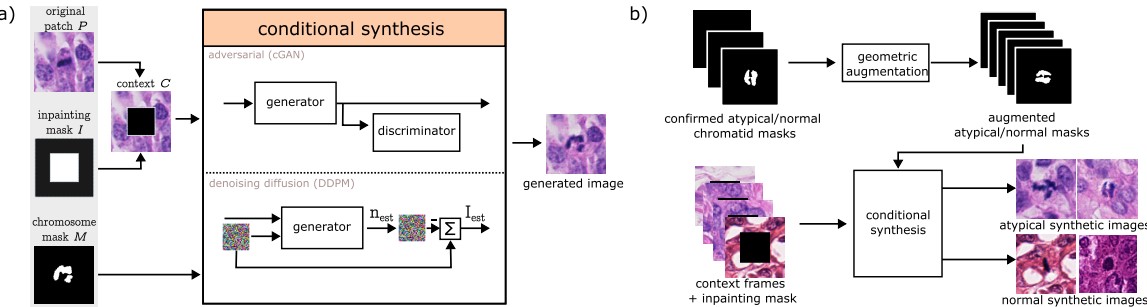

Figure 1: Overview of our approach. a) Image synthesis is conditioned on the tissue patch, inpainting mask, and chromosome mask. We evaluate both adversarial networks and denoising diffusion models. b) Image generation of synthetic atypical and normal MFs using augmented masks and cropped tissue areas.

(Kropp et al., 2024; Zhang et al., 2020). Based on the idea of leveraging inpainting to tackle class imbalance, our main contributions can be summarized as follows:

- We present a method for generation of synthetic samples to augment the problem of AMF vs. normal MF classification, employing generative inpainting models conditioned additionally on chromosome segmentation, allowing disentanglement of MF morphologic shape and tissue context.

- We adapt this approach to two generative architectures (generative adversarial networks and diffusion models) and demonstrate improved downstream classification performance on three publicly available datasets.

## 2 Materials

In this study, we utilize three public datasets for MF classification across multiple domains. All datasets consist of Hematoxylin & Eosin (H&E)-stained images including MFs. Collectively, these datasets offer a comprehensive foundation for studying MF classification across multiple domains, species, and organ types:

- The **AMi-Br** dataset (Bertram et al., 2025) comprises 3,720 MF annotations from human breast cancer samples scanned on five different scanners. These were drawn from the TUPAC challenge (Veta et al., 2019) (using the improved labels of (Bertram et al., 2020)) and the MIDOG 2021 challenge (Aubreville et al., 2023a). Of the annotations, 832 (22.36%) are AMFs by majority vote of three pathologists. We use AMi-Br for training, with a patient-level split with 41 patients (20.30%) as the **Ami-Br test** set.

- The **Atypical and Normal Mitosis (AtNorM)**-MultiDomain **(MD)** dataset (Banerjee et al., 2025) extends beyond human breast cancer and covers six domains, spanning both human and canine tumors. It was sourced from all but the human breast cancer domains of the publicly available MIDOG++ dataset (Aubreville et al., 2023b). It comprises 2,107 MFs, with 219 (10.4%) being atypical. This dataset is used as a hold-out test dataset.

- The **AtNorM-Br** (Banerjee et al., 2025) contains 746 MF instances from 179 patients, with 128 (17.16%) being atypical. This dataset is sourced from the breast cancer (BRCA) cohort of the The Cancer Genome Atlas (TCGA) (Lingle et al., 2016). TCGA contains images from various sources and with partially mixed quality and is thus a valuable asset for assessing generalization. This is also used as a hold-out test dataset.

## 3 Methods

The main goal of our method is to better cover the variability of MFs/AMFs by generating data with unseen combinations of morphology and tissue context to train more robust classification models. To achieve this, our method disentangles the morphology of AMFs, which can be understood as the shape of chromosome compartments in the H&E image, and the strongly varying surrounding (context) of the cell. These variations arise both from differences between tissue and tumor types, and from tissue preparation (Jahanifar et al., 2023), and contribute to the inherent variability in tumor specimens, which we address by generating synthetic images with realistic cellular context and diverse chromosomal shapes.

### 3.1 Chromosome Mask-Conditional Mitosis Image Synthesis

Image inpainting is typically conditioned on at least two elements (Suvorov et al., 2022): (i) a context patch from the original image and (ii) an inpainting mask, which specifies areas to be filled by inpainting. We utilize a third conditioning factor, tailored to our use case: a chromosome segmentation mask, expert-annotated by a pathologist, which highlights MF chromosomes stained by hematoxylin (dark violet/blue) (see Figure 1a).

We extract RGB patches centered on each MF and zero out pixels indicated by a binary inpainting mask, thereby hiding the target region from the model. As our key contribution, we also provide a pathologist-verified, same-resolution chromosome segmentation mask. Unlike general-purpose image inpainting, where arbitrary cutouts occur (Suvorov et al., 2022; Rombach et al., 2022), specimen images never contain such cutouts, so the inpainting region can be inferred from the zeroed context alone, obviating the need for a separate mask channel. All conditioning inputs share the same spatial dimensions. We then concatenate the RGB image and the mask along the channel axis, yielding the conditioning to our models.

### 3.2 Synthetic Data Generation

We compare two conditional image generation models: (i) a conditional generative adversarial network (cGAN) model, derived from the context encoder model (Pathak et al., 2016), and (ii) a denoising diffusion probabilistic model (DDPM) (Ho et al., 2020).

The **cGAN-based model** comprises two components: a generator and a discriminator. The generator is a U-Net-based network that synthesizes a realistic mitotic region conditioned on the input image. The discriminator, implemented as a separate convolutional neural network, is trained to classify images as either real or generated. During training, the generator aims to produce images that the discriminator cannot distinguish from real ones. The discriminator is trained on both real and generated mitotic regions to learn to distinguish between them, using a binary cross-entropy loss. This adversarial feedback helps refine the generator, encouraging it to produce increasingly realistic inpainting results. The

feedback from the discriminator is then used to improve the generator, pushing it to create more realistic and convincing inpainting results. To enhance contextual awareness and capture inconsistencies at inpainting borders, the discriminator uses dilated convolutions with dilation rates of 2 and 4 in its second-to-last and last layers, respectively, while maintaining a stride of 2 throughout all convolutional layers, which we found to be beneficial in initial experiments. Additionally, as in the original context encoder paper (Pathak et al., 2016), we use an $L_1$ loss between the original image patch and the generated image. During training, the model is trained on paired image-mask samples where each image is randomly masked in a 64×64 region, and the chromosome segmentation mask is provided as additional conditioning. This strategy ensures supervised learning with pixel-level targets and encourages realistic, mask-aware inpainting. We train both generator and discriminator for 500 epochs (batch size 8) using Adam (LR $2 \times 10^{-4}$) (Kingma and Ba, 2014) .

We further investigated a **DDPM** that learns to iteratively denoise a Gaussian noise sample to recover the missing region. We opted for pixel-space diffusion, as originally proposed by Ho et al. (2020), rather than latent diffusion models (LDMs), since the latent-space encoding and decoding may act as a bottleneck for tasks that require fine-grained accuracy in pixel space (Rombach et al., 2022). Direct pixel-space denoising preserves fine-grained spatial details, particularly crucial for AMF inpainting. At each denoising step, the model receives the noisy image, the conditioning, and an embedding of the current timestep. It then predicts the original noise component, enabling stepwise denoising. The loss function is a $L_1$ loss between the predicted and actual noise, and we train for 1000 epochs using a batch size of 8 (AdamW, LR $1 \times 10^{-4}$). Following Nichol and Dhariwal (2021), we train using 1000 steps, but only use 50 steps for inference, which we found to be of equivalent quality. We use model updates based on exponentially moving averaging, which was found a key factor in model performance for diffusion models (Karras et al., 2024).

We trained both inpainting methods using two different data sets: (i) The training split of the AMi-Br dataset (ii) and the patches sourced from a public canine mammary carcinoma (CMC) whole slide image (WSI) dataset (Aubreville et al., 2020), which includes 13,907 MF annotations. We generated chromosome segmentation masks via a semi-automated workflow: first, we manually annotated a subset of patches, then train a U-Net on those annotations, and finally have a pathology expert review and correct the model's segmentations. Although only AMi-Br provides atypical vs. normal MF labels, our mask-to-image pipeline conditions solely on the mask, so generation is label-agnostic. To synthesize the images, we randomly sampled MF context frames from the respective datasets (see Fig. 1b), and paired them with randomly selected, augmented chromosome masks from AMi-Br—choosing either atypical or normal masks to set the class of each output. As mask augmentation, we used flipping, rotation, and mild scaling and shearing. In total, we generated 26,380 synthetic images per dataset. The differences in their qualitative outputs are illustrated in Figure 2, which showcases synthetic atypical and normal MF inpainting results on both AMi-Br and CMC datasets.

### 3.3 Downstream Classification Experiment

To investigate the impact of synthetically enlarged datasets on classification, we trained an **EfficientNet-B0** baseline and two foundation models, **UNI2-h** (Chen et al., 2024) and

**Virchow2** (Zimmermann et al., 2024). The foundation models were trained via: (i) **Linear probing**, in which embeddings are extracted using the frozen foundation model and a linear classifier is trained on top, which allows us to assess how well the model's representations distinguish atypical from normal MFs, and (ii) **Low Rank Adaptation (LoRA)** (Hu et al., 2022), which is a parameter-efficient fine-tuning (PEFT) (Houlsby et al., 2019) method that freezes the original weights and inserts small trainable low-rank matrices in attention layers – reducing compute and memory needs while maintaining performance close to full fine-tuning. We compared models trained on real AMi-Br data alone versus combined with synthetic images using both inpainting methods and datasets.

We employed a 5-fold cross-validation approach for AMi-Br (80% training, 20% validation, split on patient level). Since the training dataset is imbalanced, we applied weighted random sampling to guarantee a more equitable representation of AMFs during mini-batch formation. All images were resized to $224{\times}224$ pixels, and data augmentations included random horizontal flipping, random rotation within $\pm 10$ degrees, color jittering, and random resized cropping. Model selection was performed using balanced accuracy as metric on the validation set. All models were trained on the AMi-Br train set, and tested on the AMi-Br test set, the AtNorM-Br and the AtNorM-MD dataset. We used 0.5 as threshold for the binary classification task across models.

The baseline classifier was an EfficientNet-B0 pretrained on ImageNet, with the final layer replaced by a single neuron with a Sigmoid activation function. It was optimized using binary cross-entropy loss and Adam (learning rate $10^{-3}$, weight decay $10^{-4}$), with a StepLR scheduler reducing the learning rate by 0.1 every 10 epochs. Training was performed for 100 epochs per fold.

Foundation models were trained with Adam (LR $10^{-4}$, weight decay $10^{-5}$) and cross-entropy loss, using a plateau LR scheduler (factor 0.5, patience 3, minimum LR $10^{-7}$). Training lasted up to 100 epochs (batch size 8). For linear probing, extracted features were used to train a single linear layer. For LoRA fine-tuning, we adapted pretrained models with LoRA (rank 8, scaling 16, dropout 0.3), applied to attention and MLP layers; the classification head was re-initialized and trained jointly, while the backbone remained frozen.

## 4 Results

Table 1 reports balanced accuracy for models trained with and without synthetic dataset augmentation across three test sets. For all models, we also report results when using only the real AMi-Br data for training as baselines. Balanced accuracy was used as the primary metric to address class imbalance, as it equally weighs sensitivity and specificity, which we deem relevant for a potential prognostic application.

Notably, the linear probing approach with foundation models exhibits different trends compared to other training methods, showing lower performance in general and less consistent gains from synthetic augmentation, imposed by the limited adaptation allowed by only learning a single linear layer. Excluding linear probing, augmentation with synthetic images generated by cGANs and diffusion models consistently improves performance. Across datasets, the highest balanced accuracies were achieved as follows: 0.8054 on AtNorM-MD, and 0.8099 on AtNorM-Br, both using a LoRA-tuned UNI2-h model with DDPM-based augmentation on AMi-Br; and on the AMi-Br test set, 0.853 with EfficientNet-B0 under

Table 1: Results on different test sets in five-fold cross-validation and ensembling (majority voting). All values indicate balanced accuracy scores.

| | Dataset | Inpainting Algorithm | Inpainting Dataset | AMi-Br Test Set | AtNorM-MD | AtNorM-Br |
|---|---|---|---|---|---|---|
| **EfficientNet-B0** | only real (baseline) | | | $0.755 \pm 0.021$ | $0.682 \pm 0.022$ | $0.682 \pm 0.022$ |
| | real+synthetic | cGAN | CMC | $0.831 \pm 0.014$ | $0.749 \pm 0.012$ | $0.762 \pm 0.018$ |
| | | cGAN | AMi-Br | $0.840 \pm 0.015$ | $0.756 \pm 0.015$ | $0.771 \pm 0.004$ |
| | | DDPM | CMC | $0.812 \pm 0.015$ | $0.730 \pm 0.008$ | $0.766 \pm 0.012$ |
| | | DDPM | AMi-Br | $\mathbf{0.853} \pm 0.013$ | $0.762 \pm 0.010$ | $0.782 \pm 0.015$ |
| **UNI2-h** | only real (baseline) | | | $0.6578 \pm 0.0156$ | $0.5829 \pm 0.0293$ | $0.6549 \pm 0.0155$ |
| | real+synthetic | cGAN | CMC | $0.6679 \pm 0.0131$ | $0.6006 \pm 0.0159$ | $0.6132 \pm 0.0222$ |
| | | cGAN | AMi-Br | $0.6567 \pm 0.0161$ | $0.6358 \pm 0.0206$ | $0.6533 \pm 0.0161$ |
| | | DDPM | CMC | $0.5898 \pm 0.0178$ | $0.5340 \pm 0.0108$ | $0.5550 \pm 0.0069$ |
| | | DDPM | AMi-Br | $0.6924 \pm 0.0074$ | $0.6499 \pm 0.0124$ | $0.6537 \pm 0.0153$ |
| **UNI2-h (LoRA)** | only real (baseline) | | | $0.7289 \pm 0.0196$ | $0.6974 \pm 0.0198$ | $0.7221 \pm 0.0236$ |
| | real+synthetic | cGAN | CMC | $0.8356 \pm 0.0093$ | $0.7902 \pm 0.0225$ | $0.7837 \pm 0.0130$ |
| | | cGAN | AMi-Br | $0.8356 \pm 0.0101$ | $0.7713 \pm 0.0182$ | $0.7830 \pm 0.0146$ |
| | | DDPM | CMC | $0.7896 \pm 0.0107$ | $0.7115 \pm 0.0134$ | $0.7295 \pm 0.0116$ |
| | | DDPM | AMi-Br | $0.8370 \pm 0.0064$ | $\mathbf{0.8054} \pm 0.0184$ | $\mathbf{0.8099} \pm 0.0175$ |
| **Virchow2** | only real (baseline) | | | $0.6385 \pm 0.0285$ | $0.5733 \pm 0.0173$ | $0.6349 \pm 0.0237$ |
| | real+synthetic | cGAN | CMC | $0.6619 \pm 0.0058$ | $0.6207 \pm 0.0086$ | $0.6271 \pm 0.0096$ |
| | | cGAN | AMi-Br | $0.6632 \pm 0.0092$ | $0.5490 \pm 0.0080$ | $0.6329 \pm 0.0060$ |
| | | DDPM | CMC | $0.6163 \pm 0.0085$ | $0.6210 \pm 0.0088$ | $0.5914 \pm 0.0154$ |
| | | DDPM | AMi-Br | $0.7211 \pm 0.0120$ | $0.5899 \pm 0.0076$ | $0.6618 \pm 0.0117$ |
| **Virchow2 (LoRA)** | only real (baseline) | | | $0.8165 \pm 0.0129$ | $0.7296 \pm 0.0230$ | $0.7612 \pm 0.0202$ |
| | real+synthetic | cGAN | CMC | $0.8383 \pm 0.0171$ | $0.7703 \pm 0.0171$ | $0.7814 \pm 0.0256$ |
| | | cGAN | AMi-Br | $0.8502 \pm 0.0069$ | $0.7632 \pm 0.0202$ | $0.7864 \pm 0.0221$ |
| | | DDPM | CMC | $0.8224 \pm 0.0074$ | $0.7215 \pm 0.0164$ | $0.7898 \pm 0.0137$ |
| | | DDPM | AMi-Br | $0.8549 \pm 0.0104$ | $0.7707 \pm 0.0361$ | $0.7815 \pm 0.0115$ |

the same augmentation. The on average slightly lower results on AtNorM-MD suggest that different data distributions can influence performance depending on the downstream test set. Despite this slight variability, these findings underscore the overall effectiveness of the proposed adversarial and diffusion inpainting methods on cross-dataset inference tasks, with diffusion-based approaches displaying particular robustness.

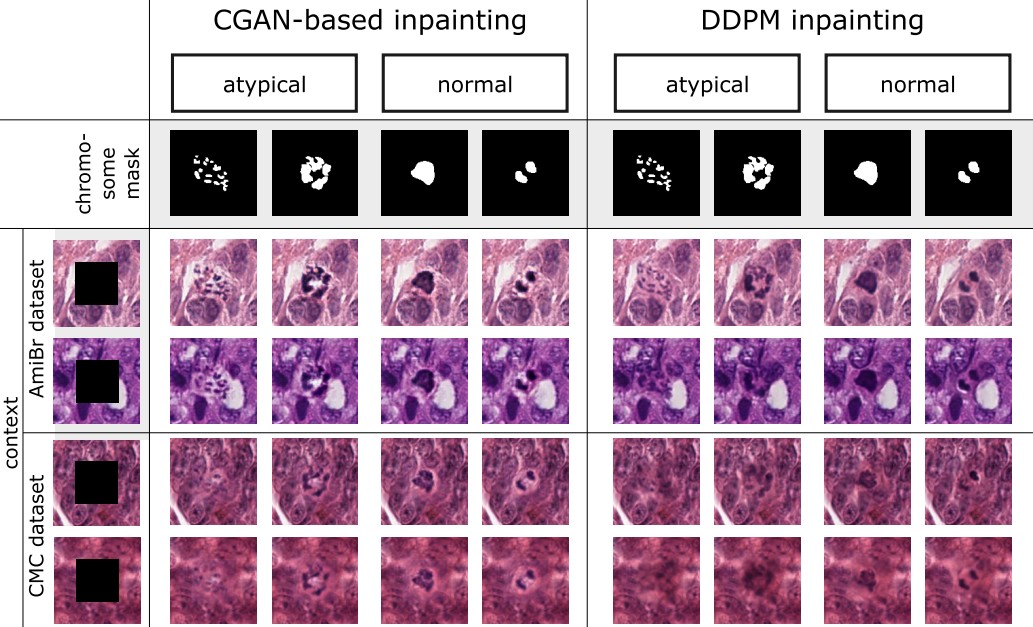

Figure 2: Exemplary results for the inpainting of atypical/normal MFs.

## 5 Discussion

In order to address the challenges in classifying AMFs in histopathology, particularly the problems of data imbalance and the scarcity of annotated samples, this paper introduces a generative strategy conditioned on three factors (chromosome masks, an image patch, and an inpainting mask). We explore this approach for both cGANs and diffusion models, demonstrating that synthetic augmentation improves generalizability across various species, tumor types, and staining conditions. While these findings suggest that integrating synthetic data can enhance automated systems for pathology review by helping pathologists identify highly prognostic AMFs more reliably, there is a limitation regarding uncertainty about whether the generated images fully reflect biological correctness. However, the consistent improvements observed across different datasets indicate that explicit expert validation, while potentially valuable for future studies, may not be strictly necessary for the effectiveness of the proposed augmentation strategy. Nevertheless, this approach lays the groundwork for balancing the atypical class in WSIs by generating synthetic AMFs, thus potentially facilitating class-balanced MF detection. Future research should consider applying these generative frameworks to other histopathological tasks where class imbalance persists, incorporating expert-in-the-loop validation to ensure the accuracy and clinical value of the synthetic images, and continuing to refine the models for greater prognostic reliability.

## Acknowledgments and Disclosure of Funding

CAB, VW, and CK acknowledge the support from the Austrian Research Fund (FWF, project number: I 6555). SB, TC, RK, and MA acknowledge support by the Deutsche

Forschungsgemeinschaft (DFG, German Research Foundation, project number: 520330054). KB and JU acknowledge support by the DFG (project numbers 405969122 and 460333672 CRC1540 EBM).

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
