# OpenReview forum: "Chromosome Mask-Conditioned Generative Inpainting for Atypical Mitosis Classification"
_MICCAI.org/2025/Workshop/COMPAYL — COMPAYL 2025_

### Official Review · Reviewer_CWc1 · 2025-07-07
**This paper introduces a novel triple-conditioned generative inpainting method for classifying atypical mitoses, which addresses data imbalance and scarcity by disentangling morphology from cellular context. While showing improved classification, it lacks rigorous biological validation of synthetic images and relies on expert-annotated masks.**

**Rating:** 4
**Confidence:** 3

**Review:**

1. Short summary: This paper introduces a novel generative inpainting strategy for classifying atypical mitoses (AMFs) in histopathology images, a task challenged by data imbalance, scarcity, and noisy labels. The proposed method utilizes a triple-conditioned generative approach, leveraging a histological context patch, an inpainting mask, and a chromosome segmentation mask to disentangle mitotic figure shape information from cellular context. The authors evaluate both adversarial (cGANs) and denoising diffusion probabilistic models (DDPMs) for image synthesis and demonstrate improved downstream classification performance for atypical vs. normal mitoses, particularly with EfficientNet-B0 and LoRA fine-tuned foundation models, across three public datasets.

2. Strengths : The triple conditioning on tissue context, inpainting mask, and chromosome segmentation mask is innovative and biologically informed, allowing for disentanglement of mitotic figure morphology from cellular context; The authors evaluate their approach across multiple datasets (AMi-Br, AtNorM-MD, AtNorM-Br) and different model architectures (GANs, diffusion models), demonstrating robustness; The authors provide complete source code, enhancing reproducibility and enabling further research.

3. Weaknesses: While the synthetic images appear visually reasonable, there's insufficient validation of their biological correctness. The authors acknowledge this limitation but don't provide expert pathologist evaluation of the generated samples;  The method requires expert-annotated chromosome segmentation masks, which may limit scalability and practical deployment. The semi-automated workflow for generating these masks is mentioned but not thoroughly evaluated; While consistent, the improvements are relatively modest (e.g., from 0.755 to 0.853 balanced accuracy on AMi-Br test set). The clinical significance of these improvements is not discussed; Linear probing with foundation models shows inconsistent and generally poor performance, suggesting the learned representations may not be optimal for this specific task.

4. Detailed Comments

The authors correctly identify the limitation regarding the biological correctness of generated images. Future work could incorporate more rigorous validation methods, perhaps involving a panel of pathologists to blindly assess the realism and biological accuracy of a larger set of synthetic images. This could provide quantitative metrics for "biological correctness.  The paper states that mask augmentation (flipping, rotation, mild scaling and shearing) was used. It would be interesting to see if different or more aggressive mask augmentation strategies yield further improvements or have any negative effects on the quality of generated images. While performance gains are clear, information regarding the training and inference times for both cGAN and DDPM models, especially in the context of generating 26,380 synthetic images per dataset, would be valuable for readers considering practical applications.

---

### Official Review · Reviewer_9Nyi · 2025-07-12
**Review of Submission 27**

**Rating:** 4
**Confidence:** 4

**Review:**

**Summary**

This paper addressed two key challenges in classifying atypical mitotic figures (AMFs) - class imbalance and data scarcity - by introducing a generative framework augmented with expert-in-the-loop validation. The method synthesizes context-aware samples. The authors compared models trained with and without synthetic data augmentation, and the results showed performance improvements across most models when synthetic samples were included.

**Strengths, weaknesses/suggestions**

Strengths:
* Clarity/quality: the paper is clearly written and engaging to read.
* Significance: while the study focused on AMF, the proposed framework could be used in broader challenges related to class imbalance in other histopathological tasks.
* The authors went through the effort of annotating chromosome, allowing the generative models to use context-aware cutouts rather than arbitrary ones.

Weaknesses/ suggestions:
* The reported performance difference between linear probing and LoRA, even in the absence of synthetic data, is interesting. A brief (theoretical) discussion would help the reader understand why linear probing (or frameworks with frozen backbone) might be less suitable for use with synthetic augmentation.
* While model improvements are clearly shown, it would be beneficial to include error analysis to suggest where synthetic data may still fall short.
* In Table 1, several metrics are very close. Reporting statistical significance or confidence intervals would clarify whether these differences are significant.

---

### Official Review · Reviewer_4Y6s · 2025-07-15
**Chromosome Mask-Conditioned Generative Inpainting for Atypical Mitosis Classification**

**Rating:** 3
**Confidence:** 4

**Review:**

The paper addresses a relevant problem and is presented in a clear way.

Weaknesses:
-The interrater disagreement is mentioned but not quantified. Since 3 raters participated in the study, it could be interesting to give some information on that, to compare with the automatic results.

-the imbalance is also mentioned but not actively tackled in the learning methodology with standard approaches but better assess the benefits of the proposal.

-"As our key contribution, we also provide a pathologist-verified, same-resolution chromosome segmentation mask." - Technically, this has already been proposed in other applications. Nevertheless, as far as I know, it's new in this specific application.
Importantly, the paper is not clear on how the loss is designed to promote the right localization of the chromosome in the generated image.

-The work uses as input bounding boxes, which are assumed 'perfectly delineated'. It would be important to assess the impact of errors in the BB localisation on the final performance.